# Approach for the Static Design of Arc-Brazed Fillet Welds from CuAl7 on Low-Alloyed Constructional Steel

**DOI:** 10.3390/ma18102339

**Published:** 2025-05-17

**Authors:** Benjamin Ripsch, Knuth-Michael Henkel

**Affiliations:** 1Fraunhofer IGP, 18059 Rostock, Germany; 2Chair of Joining Technology, University of Rostock, 18059 Rostock, Germany; knuth.henkel@uni-rostock.de

**Keywords:** arc-brazing, load-carrying fillet welds, failure criterion, CuAl7, low-alloyed structural steel, EN 1990, EN 1993, Eurocode

## Abstract

This publication covers experimental investigations on the design resistance of arc-brazed fillet welds (CuAl7) on low-alloyed structural steel (S355) subject to predominantly static loading and regarding steel construction regulations (Eurocode). In current steel construction regulations, there is no standardized design approach for arc-brazed fillet welds available, so arc-brazed connections are rarely used despite the benefits they offer in several regards compared to conventionally welded connections. Therefore, a resistance model for arc-brazed fillet welds was calibrated based on tensile tests that were conducted on gas metal arc-brazed specimens with transverse and longitudinal fillet welds. Based on the statistical evaluation of the test results according to Annex D of EN 1990, a newly determined correlation factor *β*_b_ is proposed, which can be used for the static design of arc-brazed fillet welds made of CuAl7. This approach leads to a significantly higher calculated design resistance than previous non-standardized design approaches allowed. Also, it was found that the failure behavior of the fillet welds is critical for the design resistance of the joints and that there is a need for further investigations with regard to a targeted joint failure, which, analogous to welded fillet welds, should take place along the throat of the weld and not along the less resistant diffusion zone of the joint. Thus, the results underscore the potential for the use of arc-brazed connections in steel construction in regard to their load-bearing capacity, but also highlight the necessity of continued research regarding factors influencing their structural integrity.

## 1. Introduction

Arc brazing is a thermal joining technology categorized in ISO 4063 [1] and ISO 8572 [2]. Compared to arc welding, which is the predominant joining technology in steel construction, arc brazing offers various well-known advantages due to its significantly lower energy input [3]. The process and its benefits are well established in the automobile industry, where arc brazing is used for joining pre-galvanized, low-thickness steel plates to minimize thermal distortion and zinc burn-off [4,5]. The process also offers advantages in terms of improved fatigue strength characteristics of primary load-bearing structures with attachments connected by fillet welds [6]. In view of the widespread use and productivity of gas metal arc welding (GMAW) processes in steel construction, gas metal arc brazing (GMAB) is particularly suitable for the production of arc-brazed joints on steel structures [7]. However, while there are multiple standards available covering different aspects of arc brazing like the qualification of arc-brazing processes [8] and arc-brazing personnel [9,10], destructive and non-destructive testing of arc-brazed connections [11,12], imperfections in arc-brazed joints [13], arc-brazing filler metals [14,15] and quality requirements for brazing of metallic materials [16], all of them comparable to standards applicable in steel construction, there are no standardized design rules for arc-brazed connections available that are compatible with steel construction requirements.

As demonstrated by the acquisition of construction approval for the assessment of load-carrying arc-brazed fillet welds by the holder of abZ/abG Z-30.6-76 [17], the use of arc brazing in steel construction is of interest and beneficial in cases where galvanized construction products need to be permanently connected and where bolting or other joining technologies are not an option, in cases where zinc-burn-off needs to be limited (e.g., in the inside of hollow sections), and also when low thermal distortion is desired, such as with thermally joining components with low sheet thicknesses. Another use case is cyclically loaded steel structures whose overall fatigue strength assessment can be limited by welded attachments or stiffeners. For example, bushings used for mounting different kinds of internals of wind turbine towers are welded onto the inside wall of the steel tube towers using conventional GMAW fillet welds and determine the fatigue assessment of the tower sections, as the fatigue strength of this constructional detail is lower than the fatigue strength of the main load-carrying connections (circumferential and longitudinal butt welds of the towers sections) [18]. Raising the corresponding fatigue strength for welded attachments or stiffeners from FAT80 to FAT100 or above requires the use of post-weld treatment (PWT) methods [19] and therefore the application of an additional work step in the production process. Using GMAB fillet welds instead of GMAW fillet welds would immediately result in a fatigue strength corresponding to FAT100 or above for this constructional detail [20,21,22,23] without the necessity of applying time and cost-intensive secondary work steps such as using one of the many possible PWT methods.

However, in the highly regulated field of steel construction, the use of arc brazing is still limited and dependent on the acquisition of special approvals such as [17] from building authorities, especially as no standardized design approach for arc-brazed connections has been defined to date. Thus, various well-known advantages of the low heat input thermal joining technology cannot be utilized by the construction industry without individual technology examinations and certifications from building authorities. While advantages of arc-brazed joints in terms of fatigue strength have been investigated and published for various detail categories relevant to steel construction already [6,21], the currently available investigations and approaches regarding the static design of arc-brazed fillet welds are based on a relatively small database and lead to a comparatively low static design resistance of arc-brazed fillet welds compared to regular fillet welds [17,22,23]. Current research is focused more on technological and metallurgical examinations of arc-brazing processes and arc-brazed connections [4,5,24,25,26].

Thus, the aim of this work is to reexamine the design resistance of arc-brazed fillet welds by calibrating a resistance model for GMAB fillet welds in accordance with Eurocode demands [27,28,29]. Based on tensile tests conducted on gas metal arc-brazed specimens with transverse and longitudinal fillet welds, a correlation factor *β*_b_ shall be derived from statistical evaluation of the test results according to EN 1990 (Eurocode 0) Annex D [27], which can be used for the static design of arc-brazed fillet welds.

As the weld geometry and failure behavior of the GMAB fillet welds are similar to GMAW fillet welds, the basic procedure for determining the static design resistance of fillet welds acc. to Eurocode (described below) can be adapted and a corresponding design method can be derived for arc-brazed fillet welds [17,23]. According to Eurocode 3 [29], the effective throat thickness *a* of fillet welds is usually measured from the root of the weld, located at the former surface level of the fusion faces, see Figure 1a. Consideration of a deeper penetration requires assurance that the required penetration can be consistently achieved. As stated in EN 1993-1-8 [29], the design resistance of fillet welds subject to static loading can be determined either by the directional method or by the simplified method.

In the directional method, the forces that can be transmitted per unit length of the weld are distributed parallel and transverse to the longitudinal axis of the weld and normal and transverse to the position of the plane of the weld throat (design throat area A_w_ [29]). The stresses present in the weld can be classified by stress type and stress direction as shown in Figure 2.

If the conditions according to Equations (1) and (2) are met, the design resistance of a fillet weld can be considered sufficient [29]. Normal stress parallel to the axis of the weld $\sigma_{∥}$ is considered not to be decisive on the load-bearing capacity of the weld and is therefore neglected in the calculation [30], so only normal stress perpendicular to the throat section ($\sigma_{⊥}$) and shear stress acting in the throat section perpendicular ($\tau_{⊥}$) and parallel ($\tau_{∥}$) to the axis of the weld are considered in Equation (1).(1)$$\sqrt{{\sigma_{⊥}}^{2}+3\left({\tau_{⊥}}^{2}+{\tau_{∥}}^{2}\right)}\;\leq \;\frac{f_{u}}{\beta_{w}\cdot \gamma_{M2}},$$(2)$$\sigma_{⊥}\;\leq \;{0.9\;f}_{u}/\gamma_{M2}\;,$$

Ultimate tensile strength *f*_u_ is usually determined by the weaker of the connected components [29]. The correlation factor for fillet welds *β*_w_ enables the calibration of a design function in such a way that the required safety level between the nominal resistance and the design value is maintained and not fallen short of or exceeded [30,31]. For welded joints, *β*_w_ must be selected depending on the steel grade used. The partial factor for the ultimate limit state of weld seams regarding failure due to tensile stress (resistance of cross-sections in tension to fracture) is defined as *γ*_M2_ = 1.25 [28]. The verification is carried out at the level of the tensile strength of the components, as plastic deformations within the weld seams are generally negligible for the deformation behavior of the overall structure [32].

For a fillet weld stressed exclusively transverse to the longitudinal direction of the weld, the design limit stress *σ*_w,Rd_ can be calculated according to Equation (3) [32], with *F* being the maximum axial force and *A* being the smallest throat area of the fillet weld.(3)$$\frac{f_{u}}{\beta_{w}\cdot \gamma_{M2}}={\;\sigma }_{w,\mathrm{Rd}}=\sqrt{2}\cdot \frac{F}{A}\;,$$

For fillet welds stressed exclusively parallel to the longitudinal direction of the weld, the design limit stress determined using the directional method is lower by a factor of $\sqrt{2}/\sqrt{3}$ = 0.81 than for fillet welds stressed transverse to the longitudinal direction of the weld, see Equation (4) [32].(4)$$\frac{f_{u}}{\beta_{w}\cdot \gamma_{M2}}=\sigma_{w,\mathrm{Rd}}=\sqrt{3}\cdot \frac{F}{A}\;,$$

Alternatively, to the directional method, the design resistance of a fillet weld using the simplified method may be assumed to be sufficient if the design value of the weld force per unit length *F*_w,Ed_ is not higher than the design weld resistance per unit length *F*_w,Rd_, see Equation (5) [29].(5)$$F_{w,\mathrm{Ed}}\;\leq \;F_{w,\mathrm{Rd}},$$

Regardless of the orientation of the effective fillet weld surface to the acting force, the design weld resistance *F*_w,Rd_ must be determined in accordance with Equation (6) as the product of the design shear strength of the weld *f*_vw,d_ and the effective throat thickness *a*. For long weld lengths, the design resistance must be reduced depending on the steel grade due to the uneven stress distribution along the weld [29].(6)$$F_{w,\mathrm{Rd}}=f_{\mathrm{vw},d}\cdot a,$$

The design shear strength *f*_vw,d_ of the fillet weld should be determined from the ultimate tensile strength *f*_u_ of the weaker of the base materials and the correlation factor *β*_w_ in accordance with Equation (7). *β*_w_ and *γ*_M2_ are selected in compliance with the directional method [29].(7)$$f_{\mathrm{vw},d}=\frac{f_{u}}{{\sqrt{3}\cdot \beta }_{w}\cdot \gamma_{M2}}\;,$$

For GMAB fillet welds made from CuAl7, approaches for calculating the design resistance are proposed in abZ/abG Z-30.6-76 [17] and the report to research project P 1282 [23]. In contrast to the design of regular fillet welds, both approaches use the ultimate tensile strength of the filler metal for the determination of the design shear strength of GMAB fillet welds. Technical approval abZ/abG Z-30.6-76 [17] covers an approach based on the simplified method acc. to EN 1993-1-8 [29] and prescribes the use of a correlation factor of *β*_B_ = 1.0, a partial factor of *γ*_MB_ = 1.5 and a tensile strength of *f*_u,B_ = 310 N/mm^2^. According to P 1282 [23], the design resistance can be calculated using the directional approach, using a partial factor of *γ*_M2_ = 1.25, which is in line with Eurocode recommendations, a correlation factor *β*_a_ = 1.1 and a tensile strength of *f*_u,CuAl7_ = 400 N/mm^2^. The different approaches and the resulting design resistances are summarized in Table 1.

The design resistance obtained acc. to abZ/abG Z-30.6-76 [17] is limited by the low characteristic value of tensile strength (*f*_u,B_ = 310 N/mm^2^) compared to the datasheet values for CuAl7 provided by the filler metal manufacturers (350 N/mm^2^ to 420 N/mm^2^) and the high partial factor of *γ*_MB_ = 1.5, which exceeds the Eurocode demands resistance of cross-sections in tension to fracture [29]. Also, the application of arc-brazed fillet welds is restricted to galvanized steel plates with a thickness of 1 mm ≤ *t* ≤ 3 mm, so that thicker, non-galvanized steel plates frequently used in steel construction are not covered. The design values stated in P 1282 [23] were derived from twelve tensile tests using the approach “D.7—Statistical determination of a single property” as described in EN 1990 Annex D [27]. The application of the design approach is limited to certain constructional details (bushings and round plates) and to a throat thickness of *a* = 4 mm [23]. Other methods for calculating the resistance of GMAB fillet welds, e.g., as stated in [33], are not compatible with typical regulatory design requirements at all.

For GMAW fillet welds, design resistance parameters and correlation factors are usually determined based on tests to reduce uncertainties regarding certain variables of the resistance model by calibrating the design function [27,30,32]. This way of design resistance determination is also the basis for the correlation factor values specified in EN 1993-1-8 [29,30,32,34]. Typical specimen geometries used in this context are shown schematically in Figure 3.

Considering the described state of the art regarding the design of GMAW and GMAB fillet welds, it is hypothesized that the determination of design values for GMAB fillet welds can be improved by performing tensile tests to calibrate their design function acc. to EN 1990 Annex D [27]. Furthermore, a larger sample size and a mechanized production of standard test specimens should lead to a lower scatter of test results and higher design values than previous design approaches specify. Also, the limits of existing design approaches regarding plate and throat thickness can be expanded by choosing specimen geometries that are representative of a broader field of application.

## 2. Materials and Methods

**Base metal.** Representative for a wide array of steel constructions, low-alloyed structural steel S355J2 + N acc. to EN 10025-2 [36] was chosen as base material (thickness *t* = 15 mm). The chemical composition of the base metal was determined by optical emission spectrometry (OES), see Table 2.

**Welding consumables.** As with the existing studies concerning the design resistance of GMAB fillet welds in steel construction, CuAl7 [15] was chosen as filler metal due to its relatively high material strength (*R*_m_ > 350 MPa) and good manufacturing properties when brazing unalloyed structural steels [21]. The characteristic value of tensile strength for the filler metal used was determined in [23] with *f*_uk_ = 400 N/mm^2^ (fracture strain A ≈ 40% Young’s Modulus *E* ≈ 105 GPa). An Argon-Helium process gas (I3-ArHe-70 acc. to EN 14175 [37]) was used as shielding gas.

**Specimen manufacturing.** The specimen geometry was chosen under consideration of the loading capacity of available testing machines and based on investigations to determine the design resistance of regular fillet welds [32,34]. The general specimen geometry is shown in Figure 4.

The plates were saw-cut to avoid thermal distortion and the rolling skin in the area of the fillet joints was removed by mechanical grinding. Brazing was conducted in a mechanized way using a collaborative robot (UR10 from Universal Robots GmbH, Gröbenzell, Germany) and a GMAW welding machine (S5 SpeedPulse from Lorch Schweißtechnik GmbH, Aichtal, Germany). The GMAB fillet welds were brazed continuously over the entire length of the specimens. The cut-outs on the top and bottom of the base plates as well as the waist of the specimens with transversal GMAB welds were produced by CNC milling after brazing. This way, the length of the load-carrying fillet welds *l*_TFW_ and *l*_LFW_ was adjusted and specimens with different kinds of fillet weld lengths were produced. For both kinds of specimens, half of the specimens were manufactured aiming for a throat thickness of *a* = 3 mm, and the other half was manufactured aiming for a throat thickness of 5 mm. The brazing parameters are summarized in Table 3.

Whilst wire feed rate, current *I* and voltage *U* were left constant and selected based on the visual fillet weld quality, fillet weld size and brazing consistency, an adjustment in welding speed *v*_w_ was used to manufacture fillet welds of different sizes. The difference in arc energy *E*, calculated acc. to Equation (8), is a result of the brazing parameter variation.(8)$$E=\frac{U\;\cdot \;I}{v_{w}}\;$$

After manufacturing, angular distortion and throat thickness were measured through macroscopic photographs of the specimens. Due to the limited cross-sectional visibility of the GMAB fillet welds before tensile testing, the throat thickness of the longitudinal GMAB fillet welds was measured after tensile testing using photographs of the intact part of the fillet joints located right next to the cut-out area. Figure 5 shows two manufactured specimens before testing.

Figure 6 shows the micrograph of a GMAB fillet weld cross-section. To show the comparatively small heat-affected zone in the base metal as well as the ferritic phases in the fillet weld copper matrix, the microsection was etched with Nital (3%) [38]. Towards the interface and diffusion zone of the joint, the amount of brittle and visibly darker ferritic phases inside the visibly brighter copper matrix increases, typical for arc-brazed steel copper connections [39,40].

**Tensile testing.** Force-controlled quasi-static tensile tests were performed acc. to ISO 6892-1 [41] using a servo-hydraulic testing machine (Z400E from ZwickRoell GmbH & Co. KG, Ulm, Germany). Additionally, local displacements were measured using linear variable differential transformers (LVDTs), which were placed on the ends of the stressed fillet welds using magnetic fixtures, see Figure 7. After tensile testing, the fracture surfaces of the fillet welds were documented by 3D scanning (fringe light projection using a GOM ATOS III Triple Scan from Carl Zeiss GOM Metrology GmbH, Braunschweig, Germany). The fracture surface was measured on digitally created models to determine their size as accurately as possible.

**Statistical evaluation.** The statistical evaluation of the quasi-static tensile tests was done with the aim of calibrating a resistance function in accordance with EN 1990 Annex D [27]. The basic procedure for this is shown schematically in Figure 8. The goal of the statistical evaluation is to first determine the necessity of a subdivision of the sample by evaluating the scatter of the test results. Afterwards, the distance of the design value of resistance *r*_d_, derived from the experimental test results, to the nominal resistance *r*_t,nom_, defined by the resistance function and nominal material values, is calculated, followed by calibrating the resistance function with a correlation factor, so that the required safety distance between the nominal resistance *r*_t,nom_ and the design resistance *r*_d_ is assured when using the pre-established, standardized safety factor (partial factor *γ*_M2_ = 1.25 for the design of cross-sections in tension to fracture [28]). An in-depth description of the procedure can be found in [30,31,32].

## 3. Results

### 3.1. Tensile Test Results

The results of tensile tests at specimens with longitudinal GMAB fillet welds as well as the observed failure behaviors of the specimens are shown in Figure 9a. Deviating from the expectations, the fracture of the fillet welds was not only observed to occur inside the fillet weld approximately along the theoretical throat, but also along the diffusion zone. In the stress-displacement diagram, specimens with failure along the diffusion zone are notable due to their significantly lower maximum stresses. Stresses were determined as the ratio of the tensile force during testing and the combined fracture surface area of both load-bearing fillet welds. The local displacement was determined as the mean value from the displacements as recorded by the LVDTs. Considering only the fillet welds with fractures along the throat alone, the measured fracture areas *A*_fracture_ of the longitudinal fillet welds are on average 8% higher than the fracture areas *A*_calc_ calculated from the measured throat thickness and weld length.

The results of tensile tests at specimens with transversal GMAB fillet welds are shown in Figure 9b. As with the longitudinal fillet welds, the fractures of the transversal fillet welds were observed to occur along the theoretical throat for approximately half of the specimens and along the diffusion zone for the other specimens. Again, in the stress-displacement diagram, specimens with failure along the diffusion zone are notable due to their significantly lower maximum stresses. The measured fracture areas *A*_fracture_ of the transversal fillet welds are on average 24% higher than the fracture areas *A*_calc_ calculated from the measured throat thickness and weld length, provided that only the welds with fractures along the throat are taken into account. A tabular summary of the measured and calculated surface areas as well as the tensile test results is provided in Appendix A.

### 3.2. Statistical Evaluation

#### 3.2.1. Longitudinal GMAB Fillet Welds

The results of the statistical evaluation of the tensile tests of longitudinal GMAB fillet welds are shown in Figure 10. Looking at the *r*_e_-*r*_t_ diagrams, the large scatter of the experimental values *r*_e,i_ when statistically evaluating all tensile tests together is evident. The scatter of the test results leads to a very high mean value correction factor of *b* = 3.32, even after calibration of the design function using the correlation factor *β*_b_ = 3.96, so that the design shear strength *f*_vw,d_ = 46.6 N/mm^2^ is also low. To obtain a more economical design function in line with the compatibility test acc. to Annex D of EN 1990 [27], the sample was then reduced to GMAB fillet welds with failure along the throat only. Despite the lower number of test specimens of *n* = 6 available for statistical evaluation, this step led to a significantly lower mean value correction factor of *b* = 1.197 and to a lower correlation factor of *β*_b_ = 0.85 as well, so that the calculated design shear strength *f*_vw,d_ = 216.1 N/mm^2^ is significantly higher compared to the joint statistical evaluation of all tests conducted. The inclusion of test specimens with failure in the diffusion zone in the according diagram is informative only and is based on the determination of *r*_e,i_ with the correlation factor from the reduced data set. This illustrates that the experimental values *r*_e,i_ of all test specimens with failure in the diffusion zone are lower than the corresponding theoretical values *r*_t,i_ and thus lie on the unsafe side, i.e., below the design function *r*_t_.

#### 3.2.2. Transversal GMAB Fillet Welds

The results of the statistical evaluation of the tensile tests of transverse GMAB fillet welds are shown in Figure 11. As with the longitudinal GMAB fillet welds, the large scatter of the experimental values *r*_e,i_ is evident when statistically evaluating all tensile tests together. Again, the scatter of the test results leads to a very high mean value correction factor of *b* = 1.76, even after calibration of the design function using the correlation factor *β*_b_ = 1.79, so that the design shear strength of *f*_vw,d_ = 126.6 N/mm^2^ is also low. To obtain a more economical design function in line with the compatibility test acc. to Annex D of EN 1990 [27], the sample was also reduced to GMAB fillet welds with failure along the throat only. Despite the lower number of test specimens of *n* = 9 available for statistical evaluation, this step led to a significantly lower deviation from the mean of *b* = 1.197 and a lower correlation factor of *β*_b_ = 0.85, so that the calculated design shear strength *f*_vw,d_ = 216.1 N/mm^2^ is significantly higher compared to the joint statistical evaluation of all tests conducted as well.

#### 3.2.3. Combined Statistical Evaluation

The combined statistical evaluation of the tensile tests conducted is shown in Figure 12. As with the individual statistical evaluations, correlation factor *β*_b_ is considerably lower when examining specimens with throat failure only. With *β*_b_ = 0.84, the correlation factor found by combined statistical evaluation has the lowest value determined in this study.

In Figure 13, the maximum experimental stress *F*_max_/*A*_fracture_ of the individual tests is plotted against the GMAB fillet weld length *l*_nom,i_ and against the average GMAB fillet weld thickness *a* in order to recognize dependencies regarding these two geometric fillet weld characteristics.

Regarding the fillet weld length, a tendency towards a slight decrease in the design resistance can be seen with longer welds. Also, a slight decrease in the design resistance can be seen with an increase in fillet weld thickness. Looking at Figure 13 it becomes evident that fillet weld failure along the diffusion zone occurs particularly often with fillet weld thicknesses above 4 mm. The maximum stress experimentally endured by test specimens with fractures in the diffusion zone mostly falls below the statistically determined design shear strength.

### 3.3. Fracture Surface Evaluation

#### 3.3.1. Failure Along the Throat

A failure along the throat of the GMAB fillet welds was observed in both the specimens with transverse fillet welds as well as the specimens with longitudinal fillet welds. A representative of this type of fracture, the fracture surface of a specimen with a transverse fillet weld was observed after failure using a scanning electron microscope (SEM), see Figure 14. The fracture surface is characterized by honeycombs, which are indicative of a ductile tensile fracture (honeycomb fracture) typical for bronze alloys. At the transition from GMAB fillet weld to steel base plate, a secondary crack can be seen between the base and filler material running perpendicular to the crack propagation direction.

#### 3.3.2. Failure Along the Diffusion Zone

The fracture surface of a GMAB fillet weld failure along the diffusion zone is shown in Figure 15. Diffusion zone failures occurred seemingly randomly on the upper plate flank, lower plate flank or on the same plate flank simultaneously for the longitudinal fillet welds. With the transversal welds, diffusion zone failure occurred at the bottom plate only. While most of the fillet weld volume stuck to one of the two steel base plates, a thin layer of bronze can be seen on the base plate from which the fillet weld has separated. Two different zones can be distinguished in the area of the remaining, thin bronze adhesion. Close to the former GMAB weld root, the adhering layer of filler material is apparently thinner, while the filler material adhesion near the GMAB weld toe appears to be thicker. On a microscopic level, steps running transverse to the direction of loading can be seen in the remaining filler metal. In the area with greater filler metal adhesion, the steps are more pronounced. On the surface of the filler metal steps, shear honeycombs aligned with the stress direction can be seen.

Looking at the material contrast or BSE (backscattered electrons) images of the fracture surface taken by SEM, a significantly higher proportion of oxide inclusions can be seen in the copper matrix of the thin filler metal layer close to the root compared to the thicker filler metal adhesion near the weld toe. Areas of worn fracture surfaces, characterized by a smooth surface, were more pronounced near the fillet weld toe than near the former root of the weld.

## 4. Discussion

**Directionality.** As with regular GMAW welds, the directional dependence of the design resistance of the fillet welds was already taken into account in the respective design models used for transverse and longitudinal GMAB fillet welds. Thus, the statistically determined design shear strengths were similarly high for shear and tensile loading, whereby the design resistance of longitudinal fillet welds (*f*_vw,d_ = 216 N/mm^2^) was slightly higher than that of the transverse fillet welds (*f*_vw,d_ = 204 N/mm^2^). In comparison, the design resistance of GMAW transverse fillet welds is usually higher than the design resistance of longitudinal fillet welds, even taking into account directionality [32]. The deviation from the expectation of the GMAB fillet welds can presumably be attributed to the effect of statistical evaluation of a relatively small number of tests and the greater scatter of the results on the transverse fillet weld specimens (*n* = 9) compared to the test results determined on longitudinal fillet weld specimens (*n* = 6). Correlation factors specified in Eurocode 3 apply regardless of the chosen design method, so the highest correlation factor determined needs to be considered for a conservative joint design. Therefore, the correlation factor determined on the transverse fillet weld specimens was chosen as the appropriate design value with *β*_b_ = 0.90.

**Design approach comparison.** The design shear strengths of arc-brazed and welded fillet welded joints determined using different design approaches are shown in Figure 15. Based on the new correlation factor *β*_b_, the design shear strength determined for CuAl7 within the scope of this work is significantly higher than the value of *f*_vw,d_ = 168 N/mm^2^ (+22%) determined according to research project P 1282 [23]. The difference to the design shear strength determined in accordance with abZ/aBG Z-30.6-76 [17] is even higher (+72%), see Figure 16. The causes of these differences are the differently defined characteristic tensile strength values of the filler material as well as the different correlation factors applied. In contrast to the previous design recommendations, the partial factor does not have to be increased by using the correlation factor (*β*_b_ > 1), but can even be reduced on the basis of the calibration of the design function (*β*_b_ < 1). A correlation factor < 1 means that the actual design resistance of the brazed fillet welds is higher than the theoretical design resistance when taking into account the nominal cross-sectional area of the weld, the average tensile strength of the filler metal and the safety concept of the Eurocodes (*γ*_M2_ = 1.25). In view of the high tensile strength determined in research project P 1282 [23] on tensile test specimens from single-layer CuAl7 GMAB fillet welds (*R*_m_ = 624 kN), the reduction of the partial factor due to the correlation factor does not surprise.

**Comparison to GMAW fillet welds.** The design shear strength of arc-brazed fillet welds determined by the new approach is only slightly lower than the design shear strength of GMAW fillet welds on steel with a characteristic tensile strength of *f*_u_ = 360 N/mm^2^ (S235 according to EN 20025-2 [36]), see Figure 16 as well. The reason for that is the higher reduction of the partial factor for the welded joints (*β*_w_ = 0.8) compared to the soldered joints (*β*_b_ = 0.9). Due to higher correlation factors, a higher base material strength does not lead to a proportionally higher design shear strength of GMAW fillet welds, so the design shear strength of brazed fillet welds made of CuAl7 is not more than 20% below the design shear strength of steel grades S275 to S460 in any of the cases considered. Based on the newly determined correlation factors, brazed CuAl7 fillet welds achieve calculated design resistances that are significantly closer to those of GMAW fillet welds than were possible with previous design proposals.

**Imperfections and elongation.** The angular distortion measured on the specimens (<1°) was not considered in the stress calculation and led to a conservative determination of the correlation factors, as bending stresses were thereby neglected. The low amount of elongation measured until failure (failure strain) can also be seen in tests on GMAW fillet weld specimens [32] and does not indicate a low failure strain of the filler metal itself.

**Limitations.** Since the design resistance of joints in which the fracture was located in the diffusion zone was significantly lower than that of specimens with a fracture along the throat of the fillet weld, it must be ensured that the fillet welds do not fail in the diffusion zone for the developed design model to be valid. Also, the design resistance of GMAB fillet welds in combination with different kinds of surface preparation, surface coatings or other types of filler metal must be examined separately.

**Fracture location.** Fractures in the diffusion zone were observed in fillet welds that had a greater weld thickness and were brazed with a comparatively high heat input, which is known to favor the formation of brittle phases and in turn might be the reason for the failure behavior observed. For other kinds of brazed joints, the negative effect of brittle phases on the interface strength is well examined, so different methods are examined to minimize those in order to improve the structural integrity of the joints [42,43,44]. Due to the comparable, mechanized sample preparation, an influence of sample preparation on the failure location does not seem to be too likely. As there were no obvious signs of GMAB fillet weld imperfections such as lack of fusion, the use of non-destructive testing methods for the detection of insufficient bonding might not be too promising for the detection of GMAB fillet welds that are going to fail along the diffusion zone. Thus, future research should focus on examining the actual causes of diffusion zone failure and on the development of practical approaches for diffusion zone failure prevention.

## 5. Conclusions

Based on reviewing existing design approaches for arc-brazed and welded fillet welds as well as on reviewing the development of the respective design proposals, a newly determined correlation factor *β*_b_ = 0.90 was established for arc-brazed fillet welds from CuAl7 on low-alloyed structural steel. This was conducted by performing tensile tests to calibrate a GMAB fillet weld design function acc. to Annex D of EN 1990 [27].

*β*_b_ can be applied using the existing design approach for regular fillet welds according to Eurocode 3 [29] and in combination with the characteristic value of tensile strength determined for CuAl7 in [23] (*f*_u_ = 400 N/mm^2^). Using the newly determined correlation factor, the conservatively determined design shear strength for CuAl7 fillet welds is 205 N/mm^2^ (simplified method), which is significantly higher than the design shear strength determined with previously established approaches.

However, until the cause of the inhomogeneous failure behavior observed for GMAB fillet welds has been fully understood and can be reduced to failures along the throat only, limiting the GMAB fillet weld thickness to a ≤ 4 mm as well as limiting the heat input to *E* ≤ 0.7 kJ/mm can be a practical approach to ensure the validity of the design values determined and should be considered in case of practical application. Future research should focus on investigating the technological and metallurgical reasons for failure in the diffusion zone of GMAB fillet welds, so methods to prevent this kind of failure behavior can be established. Also, future investigations should cover additional kinds of surface preparation, surface coatings and filler metals to broaden the field of application for GMAB fillet welds.

## Figures and Tables

**Figure 1 materials-18-02339-f001:**
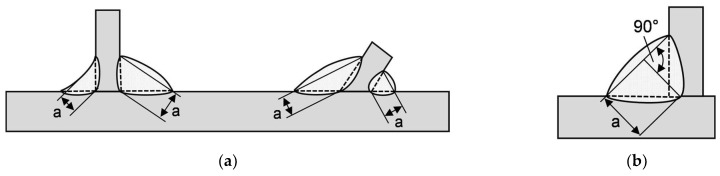
Effective throat thickness *a* of a fillet weld acc. to EN 1993-1-8 [29] (**a**) Equal and unequal leg length; (**b**) Deep penetration fillet weld.

**Figure 2 materials-18-02339-f002:**
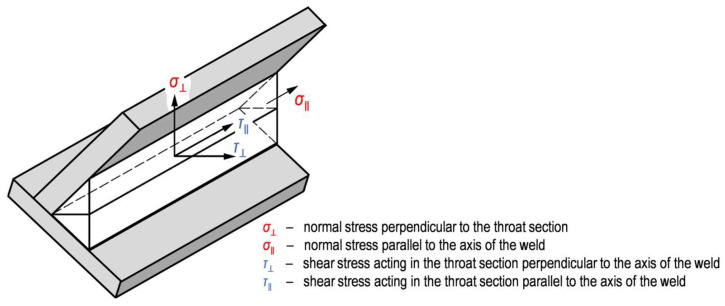
Fillet weld stresses acc. to Eurocode 3 [29].

**Figure 3 materials-18-02339-f003:**
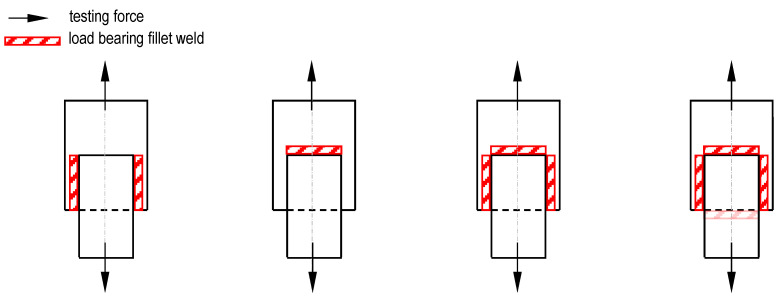
Typical specimen geometries for determining the design resistance of fillet welds acc. to [35].

**Figure 4 materials-18-02339-f004:**
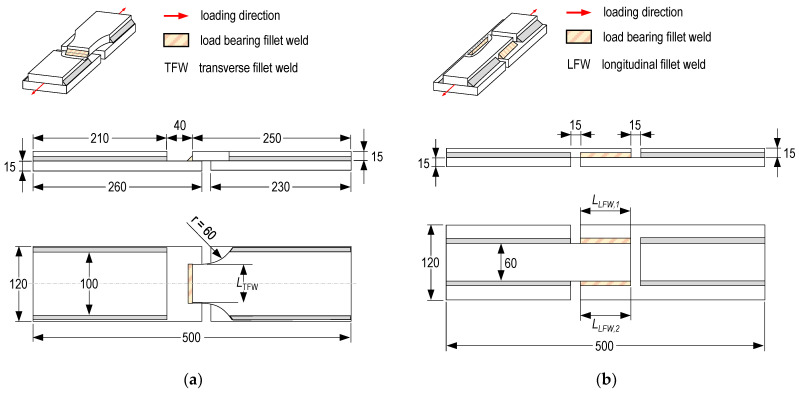
Specimen design for tensile tests of GMAB fillet welds (**a**) Transverse loading; (**b**) Longitudinal loading.

**Figure 5 materials-18-02339-f005:**
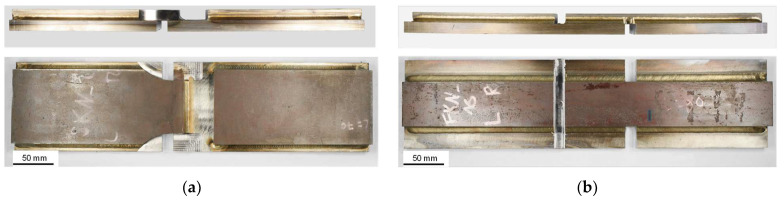
Tensile test specimens (**a**) Transversal fillet weld; (**b**) Longitudinal fillet weld.

**Figure 6 materials-18-02339-f006:**
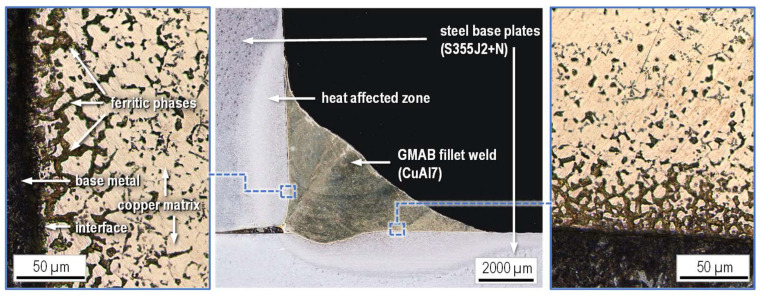
Micrograph of a CuAl7 fillet weld cross-section (etchant: Nital 3%).

**Figure 7 materials-18-02339-f007:**
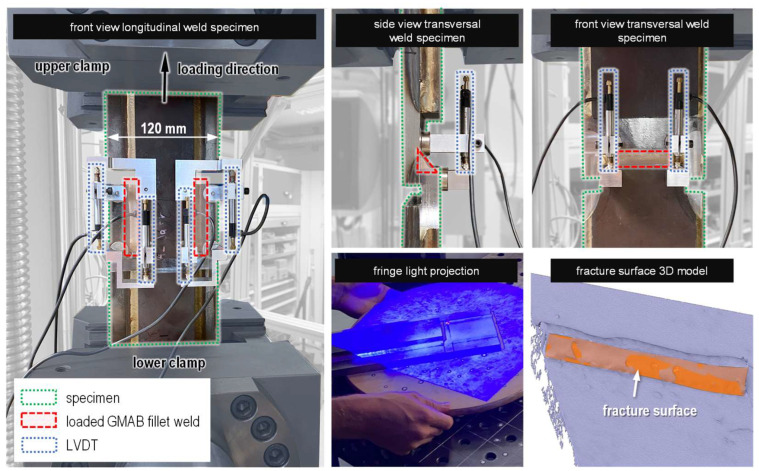
Tensile testing setup and fracture surface determination by 3D scanning.

**Figure 8 materials-18-02339-f008:**
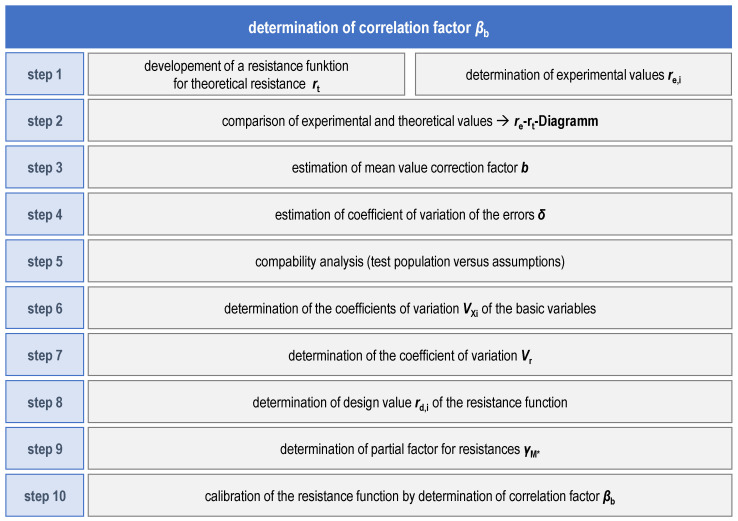
Statistical evaluation of test results for calibrating a resistance function in accordance with EN 1990 Annex D [27].

**Figure 9 materials-18-02339-f009:**
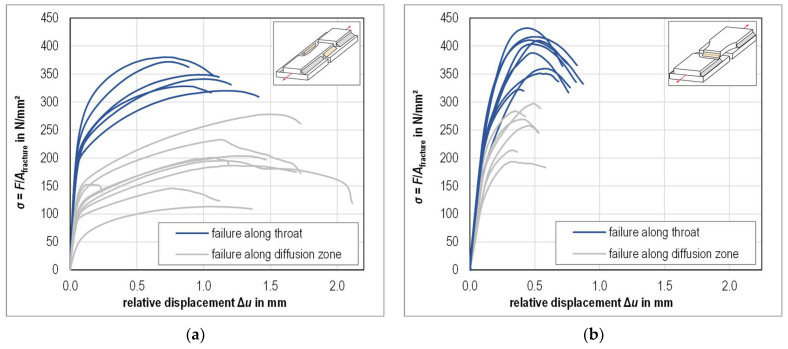
Tensile test results (**a**) Longitudinal GMAB fillet welds; (**b**) Transversal GMAB fillet welds.

**Figure 10 materials-18-02339-f010:**
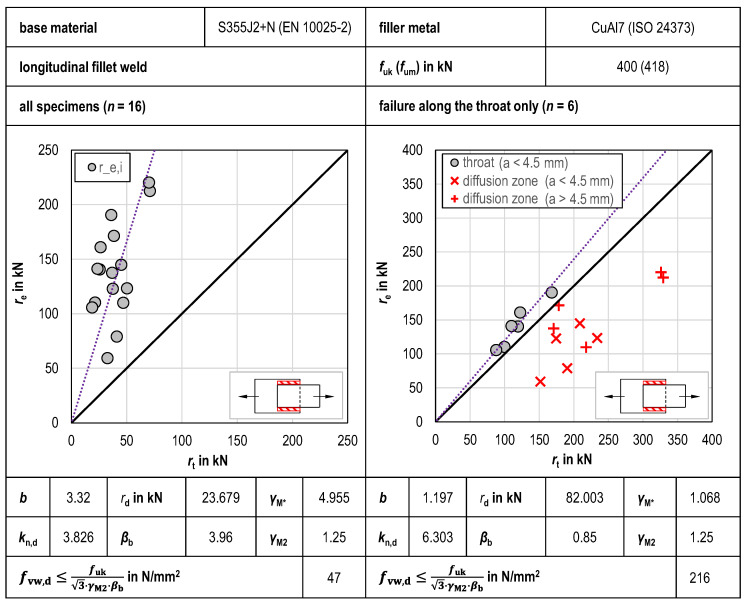
Statistical evaluation of tensile tests of longitudinal GMAB fillet welds.

**Figure 11 materials-18-02339-f011:**
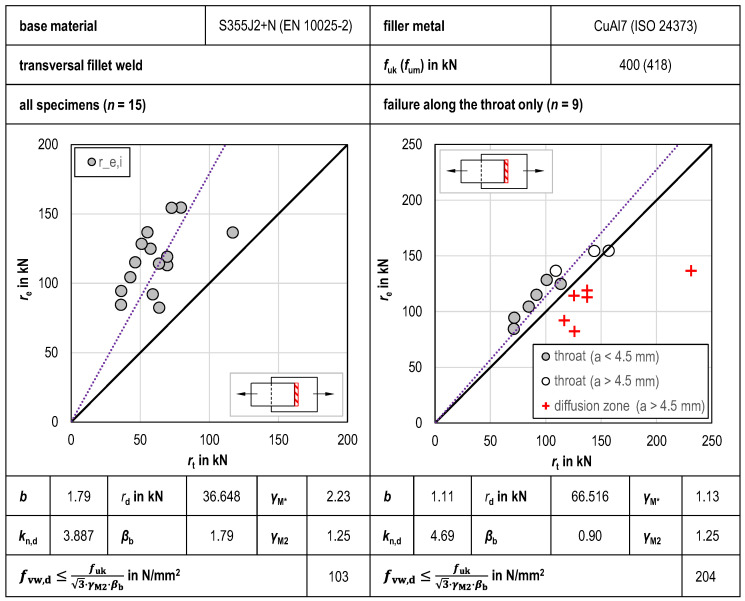
Statistical evaluation of tensile tests of transversal GMAB fillet welds.

**Figure 12 materials-18-02339-f012:**
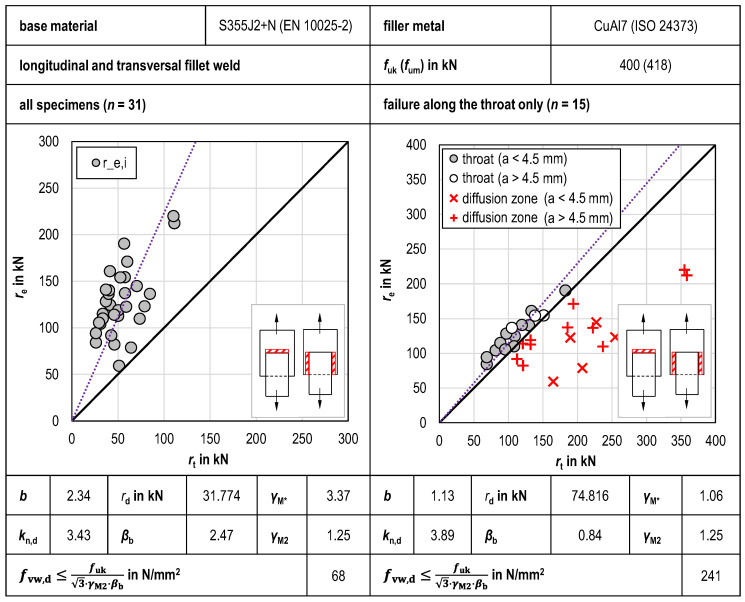
Combined statistical evaluation of tensile tests of transversal and longitudinal GMAB fillet welds.

**Figure 13 materials-18-02339-f013:**
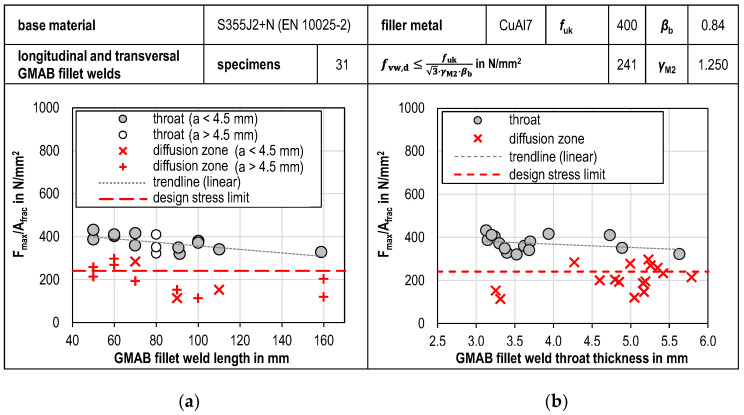
Influence of fillet weld geometry on maximum stress (**a**) Fillet weld length; (**b**) Throat thickness.

**Figure 14 materials-18-02339-f014:**
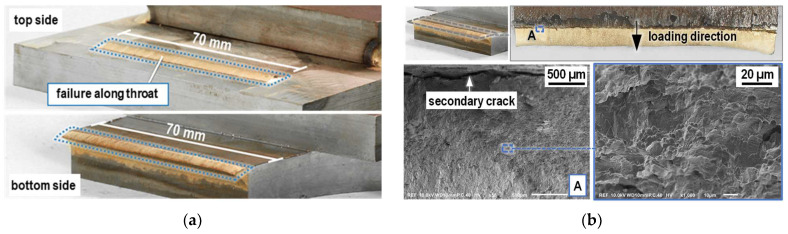
Fracture surface of a failure along the GMAB fillet weld throat (**a**) overview; (**b**) SEM image.

**Figure 15 materials-18-02339-f015:**
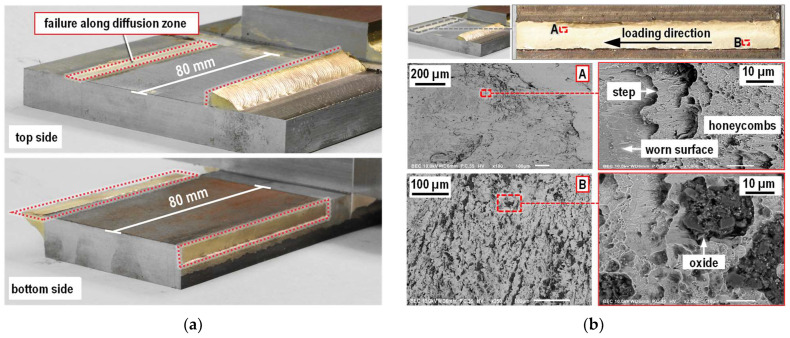
Fracture surface of a failure along the diffusion zone of a GMAB fillet weld (**a**) overview; (**b**) SEM image.

**Figure 16 materials-18-02339-f016:**
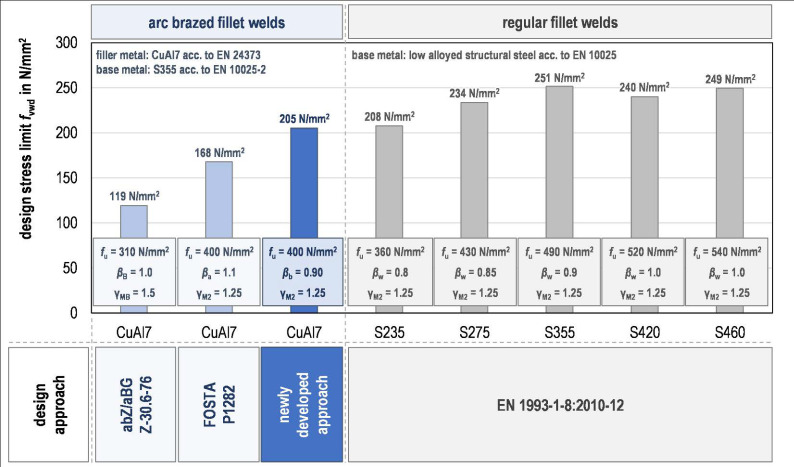
Design shear strengths of arc-brazed and welded fillet welded joints determined using different design approaches.

**Table 1 materials-18-02339-t001:** Comparison of existing design approaches for arc-brazed fillet welds.

Design Values and Design Resistance	abZ/abG Z-30.6-76 [17]	P 1282 [23]
*f*_u,B_; *f*_u,CuAl7_	310 N/mm^2^	400 N/mm^2^
*β*_B_; *β*_a_	1.0	1.1
*γ*_MB_; *γ*_M2_	1.5	1.25
design resistance of longitudinal GMAB fillet welds	119 N/mm^2^	167 N/mm^2^
design resistance of transversal GMAB fillet welds	119 N/mm^2^	205 N/mm^2^

**Table 2 materials-18-02339-t002:** Base metal composition determined by OES analysis (number of measurements *n* = 5).

Content in %	C	Si	Mn	P	S	N	Cu	Ni	Cr	Mo
$\overset{-}{x}$	0.1440	0.1880	1.320	0.0200	0.0670	0.0069	0.1730	0.0540	0.1020	0.0120
**s_x_**	0.0028	0.0032	0.014	0.0007	0.0008	0.0005	0.0026	0.0005	0.0008	0.0002

**Table 3 materials-18-02339-t003:** GMAB parameters.

Nominal Throat Thickness in mm	Wire Feed Rate in m/min	Current *I* in A	Voltage *U* in V	Welding Speed *v*_w_ in m/min	Arc Energy *E* in kJ/mm
3	5.3	170	24.4	0.4	0.54
5	5.3	170	24.4	0.18	1.36

## Data Availability

The original contributions presented in this study are included in the article. Further inquiries can be directed to the corresponding author.

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
