# Peer review of "Approach for the Static Design of Arc-Brazed Fillet Welds from CuAl7 on Low-Alloyed Constructional Steel"

_materials, 2025, doi:10.3390/ma18102339_

Round 1
Reviewer 1 Report
Comments and Suggestions for Authors
Approach for the static design of arc-brazed fillet welds from CuAl7 on low-alloyed constructional steel
The paper entitled "Approach for the static design of arc-brazed fillet welds from CuAl7 on low-alloyed constructional steel" presents an advanced and interesting approach to the study. The research equipment and innovative tools are used properly selected to realise the experimental tests. The important contribution is presented graph with the procedure of investigations, which allows tracking the next steps of Determination of correlation coefficient βb. The number of samples used in the performed experimental tests is sufficient - it allows to present of reliable statistical analyses. However, it is several questions and remarks to the presented paper:
1. Can authors explain the practical use of the presented approach for CuAl7 fillet welds for construction steel? Where the presented solution is used in the industry? The explanation of this improves the paper for readers in the context of scientific and practical significance.
2. Did the authors consider the stress concentration factor (SCF) approach to solve the presented problem?
3. In Figure 6 real testing setup is presented. The boundary conditions of the loading samples are different from those presented in the Figures 3 and 4. (fixed side and loading direction).
4. For the connection zone analyses, authors should prepare the microstructure of weld connection. This point of view and study can provide pieces of information about the cohesive zone area and possible problems with the quality of connection.
5. In Figure 8a presented results for "failure along diffusion zone". One of the sample of relative displacement is diametrically opposite (low) than others. Can authors explain that case? Can it be considered in the context of improper preparation of the sample? Can this be considered in this context? Are there any methods (NDT (non-destructive testing) for quality control) that allow for verification of this state in future analyses?
FIGURE 3 - In the image of schematic samples, the symmetry lines are needed.
FIGURE 4 - The distances between the technical drawing and dimensions should be greater, according to the technical drawing standards.
TABLE 2 - The number of decimal places after the standard deviation separator should fit the individual chemical elements mean values (for eg: P; x=0.0200; sd=0.0007). Additionally, authors should add information about the number of counts of OES analyses to calculate the mean and standard deviation.
LINE 108: The new sentence should start with the capital letter.
In future analyses for the next paper, authors can reconsider:
- try to use numerical methods (eg, Finite Element Method) to compare experimental tests with simulations;
- try to compare welding and bonding technologies to connect structural elements;
Author Response
Comment 1:
Can authors explain the practical use of the presented approach for CuAl7 fillet welds for construction steel? Where the presented solution is used in the industry? The explanation of this improves the paper for readers in the context of scientific and practical significance.
Response 1:
Another paragraph was added to the introduction, detailing practical use cases from the industry.
“As demonstrated by the acquisition of a construction approval for the assessment of load carrying arc brazed fillet welds by the holder of abZ/abG Z-30.6-76 [17], the use of arc brazing in steel construction is of interest and beneficial in cases where galvanized construction products need to be permanently connected and where bolting or other joining technologies are not an option, in cases where zinc-burn-off needs to be limited (e.g. in the inside of hollow sections), and also when low thermal distortion is desired, such as with thermally joining components with low sheet thicknesses. Another use case are cyclically loaded steel structures whose overall fatigue strength assessment can be limited by welded attachments or stiffeners. For example, bushings used for mounting different kinds of internals of wind turbine towers are welded onto the inside wall of the steel tube towers using conventional GMAW fillet welds and determine the fatigue assessment of the tower sections, as the fatigue strength of this constructional detail is lower than the fatigue strength of the main load carrying connections (circumferential and longitudinal butt welds of the towers sections) [18]. Raising the corresponding fatigue strength for welded attachments or stiffeners from FAT80 to FAT100 or above requires the use of post weld treatment (PWT) methods [19] and therefore the application of an additional work step in the production process. Using GMAB fillet welds instead of GMAW fillet welds would immediately result in a fatigue strength corresponding to FAT100 or above for this constructional detail [20-23] without the necessity of applying time and cost intensive secondary work steps such as using one of the many possible PWT methods.”
Please also note the following paragraph, which summarizes that the real-world application of GMAB in steel construction is currently highly limited due to regulatory reasons.
Comment 2:
Did the authors consider the stress concentration factor (SCF) approach to solve the presented problem?
Response 2:
The SCF approach was used in [21] and [22] to explain the beneficial fatigue strength characteristics of GMAB attachments compared to conventionally welded attachments. However, the goal of our paper was to experimentally determine the static load bearing capacity of GMAB fillet welds. Numerical investigations were not performed in this regard and will be done as part of future research concerning the topic.
Comment 3:
In Figure 6 real testing setup is presented. The boundary conditions of the loading samples are different from those presented in the Figures 3 and 4. (fixed side and loading direction).
Response 3:
Boundary conditions as shown in Figure 6 (now Figure 7) do not seem to differ from the boundary conditions as visualized in Figures 3 and 4 to us. The welds are located as indicated in the technical drawings of the specimens (Figure 4) and the plates are axially loaded as indicated in Figure 3. In case there is room for misinterpretation, please specify further what kind of difference there could be seen between the individual figures and we will be glad to adjust them accordingly.
Comment 4:
For the connection zone analyses, authors should prepare the microstructure of weld connection. This point of view and study can provide pieces of information about the cohesive zone area and possible problems with the quality of connection.
Response 4:
The micrograph originally shown in Figure 5 was now separated and shown next to new and more detailed images of the interface of the connection (Figure 6). Also, the following description was added for the newly added Figure 6:
“Figure 6 shows the micrograph of a GMAB fillet weld cross section. To show the comparatively small heat affected zone in the base metal as well as the ferritic phases in the fillet weld copper matrix, the microsection was etched with Nital (3 %) [38]. Towards the interface and diffusion zone of the joint, the amount of brittle and visibly darker ferritic phases inside the visibly brighter copper matrix increases, typical for arc brazed steel copper connections [39,40]”
The discussion benefits from the new Figure, as brittle phases along the diffusion zone are mentioned as possible reason for failure along the diffusion zone. Otherwise, no flaws/obvious problems were seen with the quality of the connection which could indicate a failure along the diffusion zone.
Comment 5:
In Figure 8a presented results for "failure along diffusion zone". One of the sample of relative displacement is diametrically opposite (low) than others. Can authors explain that case? Can it be considered in the context of improper preparation of the sample? Can this be considered in this context? Are there any methods (NDT (non-destructive testing) for quality control) that allow for verification of this state in future analyses?
Response 5:
The sample preparation was very much equal for all specimens, so even though it cannot be fully ruled out, an influence of sample preparation on the failure location does not seem to be too likely. With the diffusion zone failure of some specimens, the displacement happened on the same flanks, with others they happened on diametrically opposed flanks. At first glance the failure location seems to be random, but surely needs further investigation in future research. With NDT methods, obvious flaws like lack of fusion could be detected. However, the reason for diffusion zone failure is regarded to be metallurgical considering the correlation to heat input/throat thickness, so NDT methods might not usable for failure mode prediction.
The following sentences/paragraphs were added:
3.3.2, first paragraph, second sentence: “Diffusion zone failures occurred seemingly random on the upper plate flank, lower plate flank or on the same plates flank simultaneously for the longitudinal fillet welds. With the transversal welds, diffusion zone failure occurred at the bottom plate only.”
4., new paragraph: “Failure Location. Fractures in the diffusion zone were observed in fillet welds that had a greater weld thick-ness and were brazed with a comparatively high heat input, which is known to favor the formation of brittle phases and in turn might be the reason for the failure behavior observed. Due to the comparable, mechanized sample preparation, an influence of sample preparation on the failure location does not seem to be too likely. As there were no obvious signs of GMAB fillet weld imperfections such as lack of fusion, the use of non-destructive testing methods for the detection of insufficient bonding might not be too promising for the detection of GMAB fillet welds that are going to fail along the diffusion zone. Thus, future research should focus on examining the actual causes for diffusion zone failure and on the development of practical approaches for diffusion zone failure prevention.
Comment 6:
FIGURE 3 - In the image of schematic samples, the symmetry lines are needed.
Response 6:
Symmetry lines were added in Figure 3.
Comment 7:
FIGURE 4 - The distances between the technical drawing and dimensions should be greater, according to the technical drawing standards.
Response 7:
Distances between the technical drawing and dimensions were adjusted accordingly.
Comment 8:
TABLE 2 - The number of decimal places after the standard deviation separator should fit the individual chemical elements mean values (for eg: P; x=0.0200; sd=0.0007). Additionally, authors should add information about the number of counts of OES analyses to calculate the mean and standard deviation.
Response 8:
The number of decimal places was adjusted accordingly. Also, the number of OES measurements performed was added to the description of the table (n = 5).
Comment 9:
LINE 108: The new sentence should start with the capital letter.
Response 9:
Added “Technical approval […]” before “abZ/abG[…], so the sentence now starts with a capital letter.
Comment 10:
In future analyses for the next paper, authors can reconsider:
-
- try to use numerical methods (eg, Finite Element Method) to compare experimental tests with simulations;
- try to compare welding and bonding technologies to connect structural elements
Response 10:
Thanks a lot for the thorough review and the helpful suggestions. Your points are valid, and we tried to cover all of them to the best of our knowledge. Both of the additional, last points will be considered in future research. For the edited manuscript please see the attachment.

Reviewer 2 Report
Comments and Suggestions for Authors
The design resistance of arc-brazed fillet welds on low-alloyed structural steel subject to predominantly static loading and regarding steel construction regulations is investigated in this paper. A newly determined correlation factor is established. The failure behavior of the fillet welds is found to be critical for the design resistance of the joints. However, some mistakes in this paper need to be revised.
Some specific comments relating to parts of the paper are given as follows.
- The symbols in Equation (1) need to be consistent with the corresponding symbols in Figure 2.
- The symbols in all the equations should be explained.
- The styles of the first letters of the phrases in all the figures should be uniform.
- The underlines of the letters in Figure 4 need to be explained.
- For improving the comprehensive frame of the welding background, please consider the following citation.
Study on the evolution processes of keyhole and melt pool in different laser welding methods for dissimilar materials based on a novel numerical model, International Communications in Heat and Mass Transfer, 2025, 163: 108629
- It is better to expound the values of GMAB parameters in Table 3.
Author Response
Thanks a lot for the thorough review and the helpful suggestions. Your points are valid, and we tried to cover all of them to the best of our knowledge.
Comment 1:
The symbols in Equation (1) need to be consistent with the corresponding symbols in Figure 2.
Response 1:
The symbols on the left side of Equation (1) are consistent with the corresponding symbols in Figure 2, only that normal stress parallel to the axis of the weld is missing in Figure 2. The following sentence was added for clarification: “Normal stress parallel to the axis of the weld is considered not to be decisive on the load bearing capacity of the weld and is therefore neglected in the calculation [30], so only normal stress perpendicular to the throat section and shear stress acting in the throat section perpendicular and parallel to the axis of the weld are considered in Equation (1).”
Comment 2:
The symbols in all the equations should be explained.
Response 2:
Symbols not explained before are now also explained in the manuscript: All the stress components from Eq. (1) as well as F and A in Eq. (3) and Eq. (4)
Comment 3:
The styles of the first letters of the phrases in all the figures should be uniform.
Response 3:
The first letters of the phrases in all figures were adjusted to be uniform (no more capital letters).
Comment 4:
The underlines of the letters in Figure 4 need to be explained.
Response 4:
Underlines of the letters in Figure 4 are now removed.
Comment 5:
For improving the comprehensive frame of the welding background, please consider the following citation.
Study on the evolution processes of keyhole and melt pool in different laser welding methods for dissimilar materials based on a novel numerical model, International Communications in Heat and Mass Transfer, 2025, 163: 108629
Response 5:
Thank you for the suggestion. However, it is not clear how a numerical study on the keyhole and melt pool in different laser welding methods relates to the topic of this manuscript, which is the experimental investigation of the load bearing capacity of arc brazed fillet welds. Please elaborate where there could be a connection to our topic and how exactly our manuscript would benefit from citing the referenced paper, and we will be glad to refer to it in the appropriate passage of the text.
Comment 6:
It is better to expound the values of GMAB parameters in Table 3.
Response 6:
The following sentences and Equation (8) were added after Table 3 to expound the values of GMAB parameters:
“Whilst wire feed rate, current I and voltage U were left constant and selected based on the visual fillet weld quality, fillet weld size and brazing consistency, an adjustment in welding speed vw was used to manufacture fillet welds of different sizes. The difference in arc energy E, calculated acc. to Equation (8), is a result from the brazing parameter variation.”
E = (U * I) / v_w (8)
Reviewer 3 Report
Comments and Suggestions for Authors
Dear Authors
This article covers experimental investigations on the design resistance of arc-brazed fillet welds (CuAl7) on low-alloyed structural steel (S355) subject to predominantly static loading and regarding steel construction regulations (Eurocode). Overall, the article is relevant, however, I list my considerations below so that the publication can be carried forward.
Abstract: Add a sentence addressing the background, challenge, or problem at the beginning of the abstract. The abstract includes key findings but a concluding sentence emphasizing the impact of the proposed method could help summarize the findings more effectively.
Introduction: The authors need to complement their study with further literature reviews. Additionally, a clearer gap identification in the existing research would strengthen the justification for this study.
Materials and Methods: The materials and methods section is well explained. The section could benefit from a clear explanation of the statistical method employed. It might be worth clarification on the rationale for the chosen numbers.
Results and discussion: The results are well relevant and presented clearly, but the discussion could benefit from a deeper analysis of the failure mechanisms in the diffusion zone. Additionally, comparing the experimental results with other existing design models would help emphasize the novelty and advantages of the proposed model.
Conclusion: The conclusions are well written and relevant, but it is necessary to discuss the limitations of the study and suggest directions for future research.
References: References are appropriate, but I recommend adding more references, especially in the introduction and discussion of results.
Comments: The content of the work is interesting, but minor improvements in abstract, introduction and discussion of results and conclusion will enhance the manuscript presentation.
Author Response
Comment 1:
Abstract: Add a sentence addressing the background, challenge, or problem at the beginning of the abstract. The abstract includes key findings but a concluding sentence emphasizing the impact of the proposed method could help summarize the findings more effectively.
Response 1:
Thank you for the suggestion. The following sentence at the beginning of the abstract was altered to make the necessity of our investigations clearer.
“[In current steel construction regulations, there is no standardized design approach for arc brazed fillet welds available], so arc brazed connections are rarely used despite the benefits they offer in several regards compared to conventionally welded connections. [Therefore, a resistance model for arc-brazed fillet welds was calibrated based on tensile tests that were conducted on gas metal arc-brazed specimens with transverse and longitudinal fillet welds.]”
Also, a concluding sentence to emphasize the impact of the proposed method was added as well:
“Thus, the results underscore the potential for the use of arc brazed connections in steel construction in regard to their load bearing capacity, but also highlight the necessity of continued research regarding factors influencing their structural integrity.”
Comment 2:
Introduction: The authors need to complement their study with further literature reviews. Additionally, a clearer gap identification in the existing research would strengthen the justification for this study.
Response 2:
A summery regarding the most relevant, existing standards covering arc brazing and arc brazed connections was added to emphasize the current lack of standardization for the assessment/design of arc brazed connections. Standards added to the introduction are listed in the following:
DIN EN 12797
DIN EN 12799
DIN EN ISO 4063
DIN EN ISO 13585
DIN EN ISO 17672
DIN EN ISO 17779
DIN EN ISO 18279
DIN EN ISO 22688
DIN ISO 857-2
ISO 11745
It was also added that current research is focused more on technological and metallurgical examinations of arc brazing processes, citing three additional papers (Pfeifer 2015, Mirski 2018, Mirski 2018), so that due to different manufacturer needs additional research is necessary regarding the assessment of arc brazed connections.
Comment 3:
Materials and Methods: The materials and methods section is well explained. The section could benefit from a clear explanation of the statistical method employed. It might be worth clarification on the rationale for the chosen numbers.
Response 3:
An abbreviated description of the basic steps and goals of the statistical evaluation was added, see below. However, it is not possible to fully explain the whole process as part of the paper, as it is explained over multiple pages in detail in EN 1990 Annex D and in the other referenced documents.
“The goal of the statistical evaluation is to firstly determine the necessity of a subdivision of the sample by evaluating the scatter of the test results. Afterwards, the distance of the design value of resistance rd, derived from the experimental test results, to the nominal resistance rt,nom, defined by the resistance function and nominal material values, is calculated, followed by calibrating the resistance function with a correlation factor, so that the required safety distance between the nominal resistance rt,nom and the design resistance rd is assured when using the pre-established, standardized safety factor (partial factor γM2 = 1.25 for the design of cross-sections in tension to fracture [27]).”
Comment 4:
Results and discussion: The results are well relevant and presented clearly, but the discussion could benefit from a deeper analysis of the failure mechanisms in the diffusion zone. Additionally, comparing the experimental results with other existing design models would help emphasize the novelty and advantages of the proposed model.
Response 4:
The possible influence of sample preparation on the failure mechanism as well as the possibility of failure mechanism prognosis by non-destructive testing was added to the discussion (both not very likely). Even though we share your interest in investigating the failure mechanism further, it shall be noted that this is the first time that the two different failure mechanism are publicly described and documented at all, which should be a benefit for the welding/arc brazing/steel construction community even without a full understanding of the underlying technological or metallurgical reasons. A more in-depth analysis of the failure mechanism requires additional experimental investigations, which cannot be provided for this article, but will be done as part of future research.
Also, in the discussion there is a designated paragraph for the comparison of the newly proposed model to the two different existing models for arc brazed joints, highlighting the benefit in terms of design shear strength obtainable using the newly determined correlation factor βb and comparing the effect of different values of the correlation factors. Furthermore, the comparison is visualized in Figure 15. Additionally, the design shear strength of arc brazed joints acc. to the new model is compared to the design shear strength of traditional welds as well, so they can be judged regarding their sufficiency for steel construction.
Comment 5:
Conclusion: The conclusions are well written and relevant, but it is necessary to discuss the limitations of the study and suggest directions for future research.
Response 5:
The limitations of the study, as discussed in section four of the manuscript, are now included in the discussion as well and they are used for suggesting directions for future research:
“Future research should focus on investigating the technological and metallurgical reasons for failure in the diffusion zone of GMAB fillet welds, so methods to prevent this kind of failure behavior can be established. Also, future investigations should cover additional kinds of surface preparation, surface coatings and filler metals to broaden the field of application for GMAB fillet welds.“
Comment 6:
References: References are appropriate, but I recommend adding more references, especially in the introduction and discussion of results.
Response 6:
Overall, 22 references were added to the introduction, discussion and other sections of the manuscript to illustrate the topic and arguments made more in depth (different standards, some books and also more research papers were added).
Comment 7:
The content of the work is interesting, but minor improvements in abstract, introduction and discussion of results and conclusion will enhance the manuscript presentation.
Response 7:
Thank you for your helpful suggestions. Your points are valid, and we covered all of them to the best of our knowledge. We hope that the additions to the abstract, introduction, discussion and conclusion meet your expectations.
Round 2
Reviewer 1 Report
Comments and Suggestions for Authors
The authors have addressed the comments properly. I suggest to continue the process of publishing of this manuscript.
Author Response
Thank you for your review and final approval of the revised manuscript.
Reviewer 2 Report
Comments and Suggestions for Authors
The authors haven’t revised the manuscript according to the comments. This paper can not be accepted.
Author Response
Dear Reviewer 2,
thanks again for your time and effort to review our paper. We did our best to revise the manuscript according to your comments from review round 1. Also, we incorporated changes and additions suggested by two other reviewers as well.
As you state that we haven’t revised the manuscript according to your comments, could you please indicate which of your comments from review round 1 you consider not to have been taken into account? In our response to Review Round 1, we responded in detail to each of your much appreciated and helpful comments.
Best regards,
Benjamin Ripsch